# A bioinspired and biocompatible *ortho*-sulfiliminyl phenol synthesis

Feng Xiong[1], Liang Lu[1], Tian-Yu Sun[2], Qian Wu[2], Dingyuan Yan[1], Ying Chen[2], Xinhao Zhang[2], Wei Wei[1], Yi Lu[1], Wei-Yin Sun[1], Jie Jack Li[3] & Jing Zhao[1,2]

Synthetic methods inspired by Nature often offer unique advantages including mild conditions and biocompatibility with aqueous media. Inspired by an ergothioneine biosynthesis protein EgtB, a mononuclear non-haem iron enzyme capable of catalysing the C–S bond formation and sulfoxidation, herein, we discovered a mild and metal-free C–H sulfenylation/intramolecular rearrangement cascade reaction employing an internally oxidizing O–N bond as a directing group. Our strategy accommodates a variety of oxyamines with good site selectivity and intrinsic oxidative properties. Combining an O–N bond with an X–S bond generates a C–S bond and an S=N bond rapidly. The newly discovered cascade reaction showed excellent chemoselectivity and a wide substrate scope for both oxyamines and sulfenylation reagents. We demonstrated the biocompatibility of the C–S bond coupling reaction by applying a coumarin-based fluorogenic probe in bacterial lysates. Finally, the C–S bond coupling reaction enabled the first fluorogenic formation of phospholipids, which self-assembled to fluorescent vesicles *in situ*.

[1] State Key Laboratory of Coordination Chemistry, Institute of Chemistry and BioMedical Sciences, School of Chemistry and Chemical Engineering, Nanjing University, Nanjing 210093, China. [2] Guangdong Key Laboratory of Nano-Micro Material Research, School of Chemical Biology and Biotechnology, Peking University Shenzhen Graduate School, Shenzhen 518055, China. [3] Department of Chemistry, University of San Francisco, 2130 Fulton Street, San Francisco, California 94117, USA. Correspondence and requests for materials should be addressed to J.Z. (email: jingzhao@nju.edu.cn).

Enzymatic C–S bond formation is a common process in biological system[1–5]. For example, ergothioneine is considered as a protectant against oxidative stress[6,7]. The key step in its biosynthesis pathway is the mononuclear non-haem iron enzyme EgtB-catalysed sulfenylation formation between γ-glutamyl cysteine and N-α-trimethyl histidine, involving a sulfur transfer step and an oxygen transfer step (Fig. 1a)[8,9].

A variety of synthetic methods have been developed to construct the *ortho*-functionalized phenols which are highly useful in chemical industry[10], functional materials[11] and medicines[12–14]. These methods mainly include three kinds of strategies: (a) rearrangement of aromatic O–X bonds[15–20]; (b) directing group-assisted *ortho* C–H hydroxylation of arenes[21–27]; and (c) *ortho* C–H functionalization of phenols[28–32]. Although these results have promoted the development of the phenol chemistry, the more efficient, economical and biocompatible methods are still in demand.

Inspired by the sulfur transferases and our previous successes in O–N bond-directed synthesis of *ortho*-functionalized phenol[33–35], we envisioned that *ortho*-sulfiliminyl phenols could be obtained by combining a directing group containing an internally oxidizing O–N bond with a sulfenylation reagent[36,37]. The desired sulfenylation reagent and oxidizing X–N bond needs to accomplish the following two tasks (Fig. 1b): (i) sulfur transfer[38,39]. A well-chosen electrophilic sulfenylation reagent would facilitate the N-sulfenylation of the X–N moiety and lead to the formation of an N–S bond to produce intermediate **B**; (ii) rearrangement. Pivotal progress was made by Maulide[40,41], Procter[32,42], Yorimitsu[31] and Peng[43] who pioneered the directed, metal-free, redox-neutral and *ortho*-functionalization. These inspiring work suggested that when the substrate captured a suitable partner, the resulting intermediate may undergo a sigmatropic rearrangement and rearomatization to product **D**, leading to the formation of a C–X bond with concurrent O–X bond cleavage. Herein, we report a rationally designed and metal-free coupling method to synthesize sulfilimines via an internal oxidant-directing strategy for the cascade formation of C–S and S＝N bonds at room temperature.

## Results

**Optimization of the reaction conditions**. For direct coupling reactions, especially those catalysed by transition metals, a directing group typically escorts the metal catalyst towards the neighbouring *ortho*-position and dictates the site selectivity. Directing groups containing N–N bond, S–N bond or O–N bond are redox versatile and could facilitate inter- or intramolecular cyclization[44–47]. At the outset of this study, compounds **1** with those bonds were firstly screened to couple with a thionating reagent N-ethylthiophthalimide **2a** under previously reported metal catalysed conditions[48–50] for similar reactions (Fig. 2a). Attempts on substrate **1** with X of N or S yielded no reaction. Gratifyingly, when X was replaced by O, the resulting N-phenoxyacetamide **1a** concurrently constructed a C–S bond and an S＝N bond, giving the desired phenolic sulfilimine product **3aa** in 83% yield.

The N–H bond in the O–NHAc moiety was found to be essential for the reaction as no reaction occurred when N–H was methylated (Fig. 2a). The need for an electron-donating phenoxy group as well as an N–H led us to suspect the existence of an ammonium ion as an essential intermediate in promoting the cascade reaction. Therefore, we removed the Rh catalyst and N₂ protection from the reaction system and the reaction could occur smoothly under metal-free conditions. Next, different sulfenylation reagents were screened to explore the cascade strategy (Fig. 2b). Tolyl sulfides with different leaving groups on

the S-atom such as chloride, tosyl and phthalimidoyl coupled with N-phenoxyacetamide **1a** to afford **3af** in 18, 33 and 85% yield, respectively. With benzenesulfenyl as the leaving group, however, no reaction took place, suggesting that disulfide remains intact during the course of the coupling reaction. As the coupling reaction was most likely mediated by a base, we tested various bases such as Et₃N, DIPEA, DBU, K₂CO₃, Na₂CO₃, NaOAc and CsOAc, where CsOAc gave the highest yield. Switching the reaction solvent to methanol and using an air atmosphere, the yield of the phenol product **3aa** was further improved to 92% (Supplementary Information, Supplementary Table 6).

**Substrate scope of the reaction**. To probe the scope of the transition metal-free cascade C–S and S＝N bond formation, we examined a series of oxyamide substrates (Table 1). Replacing the acetyl group with a bulkier pivaloyl or a benzoyl group only slightly decreased the yield to 80% (**3ba**) and 83% (**3ca**), respectively. It is worth noting that the sulfilimine substitution occurred exclusively at the *ortho*-position of the phenoxyamide moiety instead of the benzamide moiety (**3ca**), which indicated the stronger directing ability of the oxyamide group for sulfenylation. Substitutions on the phenoxy side of **1** had little impact on the yield. Electron-donating groups (**3da, 3ea, 3ia, 3la**), electron-withdrawing groups (**3ha**), as well as halogen groups (**3fa, 3ga**) were well tolerated, which afforded substituted sulfilimines in 85% to 92% yield. The C–S bond formation proceeded exclusively at the site *ortho* to the acetylaminoxy group. Therefore, for substrate **1** with two different *ortho*-sites, two regioisomers with ratio almost 1:1 were produced (**3ja:3ja′**, **3ka:3ka′**, **3ma:3ma′**, **3na:3na′**). Fusion of a benzene ring as in the substrate of naphthalene did not affect the reaction yield but resulted in high regioselectivity, which only functionalized the *ortho* C–H at C-1 position, resulting in a 2-naphthol derivative (**3oa**).

Under optimal conditions, we explored the substrate scope for N-substituted phthalimides (Table 2). The reaction proceeded smoothly for both aliphatic and aromatic thiophthalimides. Aliphatic groups including trifluoromethyl, linear alkyl and cyclic alkyl gave high yields (**3ab–3ad**, 76–92%). For aromatic thiophthalimides, substitutions on the phenyl ring increased the reaction yield (**3af–3aj** > **3ae**). The reaction proceeded well with either electron-donating groups or halogen-containing substrates.

**Synthetic application**. To further explore the applicability of our method as a useful tool in chemical biology, we conducted the reaction in PBS buffer in air. Gratifyingly, the reaction proceeded well. When the ratio of DMSO to pH 7.4 1× PBS buffer was 1:19, 81% yield was obtained (Fig. 3a, entry 1). Because of the excellent chemoselectivity under the mild aqueous conditions, we tested the compatibility of the C–S bond coupling reaction with various biomolecules, such as amino acids and proteins. The addition of a stoichiometric amount of amino acids or proteins in standard aqueous conditions did not significantly affect the reaction (Fig. 3a, entry 2~5). Bacterial cell lysates that contained various endogenous biomolecules were also tested and gave product **3aa** in 73% yield (Fig. 3a, entry 6). When we started from a non-fluorescent coumarin substrate (**1p**) to react with **2a** under such biomimetic conditions, a fluorescent turn-on process was observed. The fluorescent product **3pa** ($\lambda_{ex/em} = 360/450$ nm) was obtained in 80% yield (Fig. 3b).

Finally, we further applied the C–S bond coupling reaction to the first fluorogenic formation of phospholipids. We designed a non-fluorescent coumarin-functionalized analogue of the lysolipid 1-palmitoyl-sn-glycero-3-phosphocholine **1q** and a linear alkyl sulfenylation reagent **2k**. Phospholipids, which are the major

**Figure 1 | Strategy for the formation of *ortho*-sulfiliminyl phenol derivatives.** (**a**) The mononuclear non-haem iron enzyme EgtB-catalysed sulfenylation formation between γ-glutamyl cysteine and *N*-α-trimethyl histidine. (**b**) A metal-free approach to ortho-sulfiliminyl phenol via the C–H sulfenylation/intramolecular rearrangement cascade reaction.

**Figure 2 | Screening of the X–N functional groups and thiolating reagents.** (**a**) Screening of the multifunctional X–N functional group; reaction conditions: 0.2 mmol substrate **1**, *N*-ethylthiophthalimides (1.2 equiv.), [Cp*RhCl₂]₂ (5 mol %) and CsOAc (0.3 equiv.) in CH₃CN (1 ml) at room temperature under N₂ for 15 h. (**b**) Screening of different thiolating reagents with *N*-phenoxyamides. Reaction conditions: 0.2 mmol substrate **1a**, **2** (1.2 equiv.) and CsOAc (0.3 equiv.) in MeOH (1 ml) at room temperature for 15 h. Yields are those of isolated products. N.R. = No reaction.

component of cell membranes, have many important applications such as drug delivery[51,52], construction of micro-reactors[53] and study of protein–membrane interactions[54]. Pioneered by Devaraj *et al.*, it has been of increasing significance to develop methods for the *de novo* synthesis and assembly of phospholipid membranes[55–58]. To apply our mild C–S bond coupling reaction to the formation of the lipid vesicle under optimal conditions, we simply mixed compounds **1q** and **2k** in 0.1 M PBS buffer at pH 7.4 and sonicated the mixture at room temperature for 1 h. Blue fluorescent lipid vesicles were observed by the fluorescence microscopy after 3 h at 37 °C (Fig. 4c). We confirmed these vesicles were lipid membrane structures by staining with the membrane-

staining dye 1,1′-dioctadecyl-3,3,3′,3′-tetramethylindocarbocyanine perchlorate (DiI), and the orange red fluorescent vesicles were observed, suggesting that fluorescent phospholipid **3qk** vesicles are lipid membranes (Fig. 4c).

**Mechanistic investigation.** A combined experimental/computational study was conducted to investigate the reaction mechanism. The cross-over experiment was carried out using a 1:1 mixture of *N*-phenoxyacetamide **1a** and its analogue **1a**-*d₈* under the standard conditions, only the intramolecular rearrangement products **3aa** and **3aa**-*d₇* were obtained (Supplementary

## Table 1 | Substrate scope of aryloxyamides*.

**3aa** 92%  CDCC 1041436  **3ba** 80%  **3ca** 83%  **3da** 90%

**3ea** 90%  **3fa** 85%  **3ga** 86%  **3ha** 89%  **3ia** 89%

**3ja:3ja'** 89% (1.1:1)  **3ka:3ka'** 84% (1.05:1)  **3la** 88%

**3ma:3ma'** 85% (1:1)  **3na:3na'** 86% (1:1)  **3oa** 90%

*Reaction conditions: 0.2 mmol oxyamide, N-ethylthiophthalimides (1.2 equiv.) and CsOAc (0.5 equiv.) in MeOH (1 ml) at room temperature for 3 h. Yields are those of isolated products.

## Table 2 | Substrate scope of *N*-substituted thiophthalimides*.

**3ab** 92%  CCDC 983618  **3ac** 85%  **3ad** 76%  **3ae** 67%

**3af** 85%  **3ag** 80%  **3ah** 75%  **3ai** 70%  **3aj** 68%

*Reaction conditions: 0.2 mmol **1a**, N-substituted thiophthalimides (1.2 equiv.) and CsOAc (0.5 equiv.) in MeOH (1 ml) at room temperature for 3 h. Yields are those of isolated products.

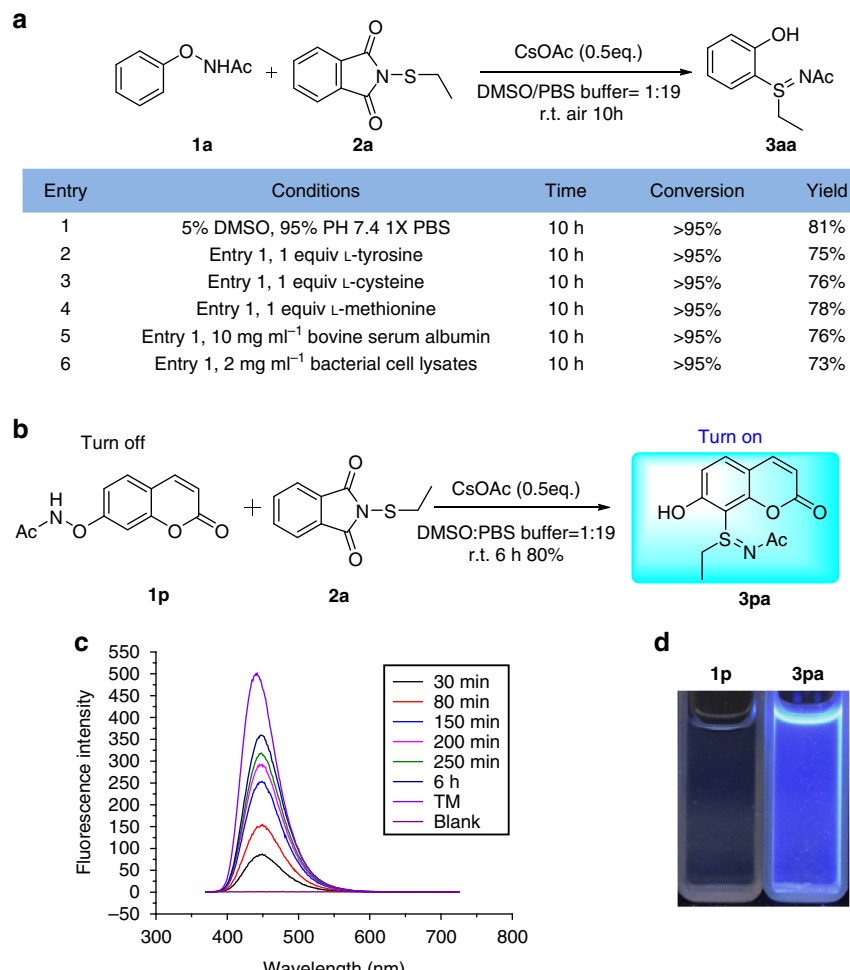

**Figure 3 | Application of the C–S bond coupling reaction in biocompatible conditions.** (**a**) C–S bond coupling reaction in aqueous conditions and in the presence of biomolecules. Conditions: **1a** (0.075 mmol), **2a** (0.09 mmol), CsOAc (0.5 equiv.), DMSO/PBS buffer = 1:19 (5 ml); the yield was determined by $^1$H NMR spectroscopy using 1,4-dimethoxybenzene as an internal standard. The temperature was RT. (**b**) Reaction of **1p** with **2a** in aqueous conditions. Conditions: **1p** (0.2 mmol), **2a** (0.24 mmol), CsOAc (0.5 equiv.), DMSO/PBS buffer = 1:19 (10 ml); isolated yield. The temperature was RT. (**c**) Fluorescence spectra of reaction **b** in aqueous conditions. (**d**) Photograph showing the visual fluorescence of **1p** and **3pa** under a 365 nm ultraviolet lamp.

Fig. 36a), suggesting an intramolecular process. The cross-over experiment between **1c**, **1d** and **2a** confirmed this conclusion (Supplementary Fig. 36b). To further probe the reaction mechanism, the potential energy surface of the proposed pathway was calculated with density functional theory. The computational results suggested that the reaction proceeds through *N*-sulfeny-lation, [2, 3] sigmatropic rearrangement and aromatization (Supplementary Fig. 37a).

In summary, we have developed a bioinspired strategy for the synthesis of *ortho*-sulfiliminyl phenols by internal oxidation-induced sulfur transfer under mild conditions. This efficient method enabled the simultaneous construction of C–S and S=N bonds. Thanks to the mild nature and good functionality tolerance of the reaction conditions, a wide range of oxyacetamides was converted into the corresponding phenols. For the sulfur donors, not only trifluoromethylthio group (CF$_3$S–) but also a variety of sulfur-containing groups were able to participate in C–H sulfenylation. The sulfur donors included *N*-substituted thiophthalimides with *S*-substituted aromatic and aliphatic groups. Moreover, the method utilized the leaving acetamide moiety of the internal oxidant/directing oxyacetamide group to construct a sulfilimine functional group. Our method was successfully applied to the *in situ* formation of fluorogenic phospholipid membranes. To the best of our knowledge, this is

the first fluorogenic phospholipid membranes formation. Further applications of the fluorogenic phospholipid membranes are under investigation and will be reported in due course.

## Methods

**Materials.** For NMR spectra of compounds in this manuscript, see Supplementary Figs 1–32. For the crystallographic data of compound **3aa** and **3ab**, see Supplementary Figs 33 and 34 and Supplementary Tables 1–5. For the representative experimental procedures and analytic data of compounds synthesized, see Supplementary Methods.

**General procedure of C–S bond coupling reaction.** Aryloxyamide (**1**) (0.2 mmol), *N*-substituted thiophthalimides (**2**) (0.24 mmol) and CsOAc (0.06 mmol or 0.10 mol) were weighed into a 10 ml pressure tube, to which was added MeOH (1 ml). The reaction vessel was stirred at room temperature for 3 h in air. Then the mixture was concentrated under vacuum and the residue was purified by column chromatography on silica gel with a gradient eluent of petroleum ether and ethyl acetate to afford the corresponding product.

***In situ* self-assembly of fluorescent vesicles.** An aliquot of 10.0 μl of a 4 mM coumarin-functionalized analogue of the lysolipid 1-palmitoyl-sn-glycero-3-phos-phocholine **1q** solution in 100 mM PBS buffer pH 7.4 was added to 2.0 μl of a 20 mM solution of sulfenylation reagent **2k** in CHCl$_3$. Then, 28 μl of a 100 mM PBS buffer pH 7.4 solution was added, and the mixture was sonicated at room temperature (RT) for 1 h. after 3 h standing at 37 °C, stained with membrane-staining dye DiI, 10 min later, the corresponding mixture was observed by fluorescence microscopy.

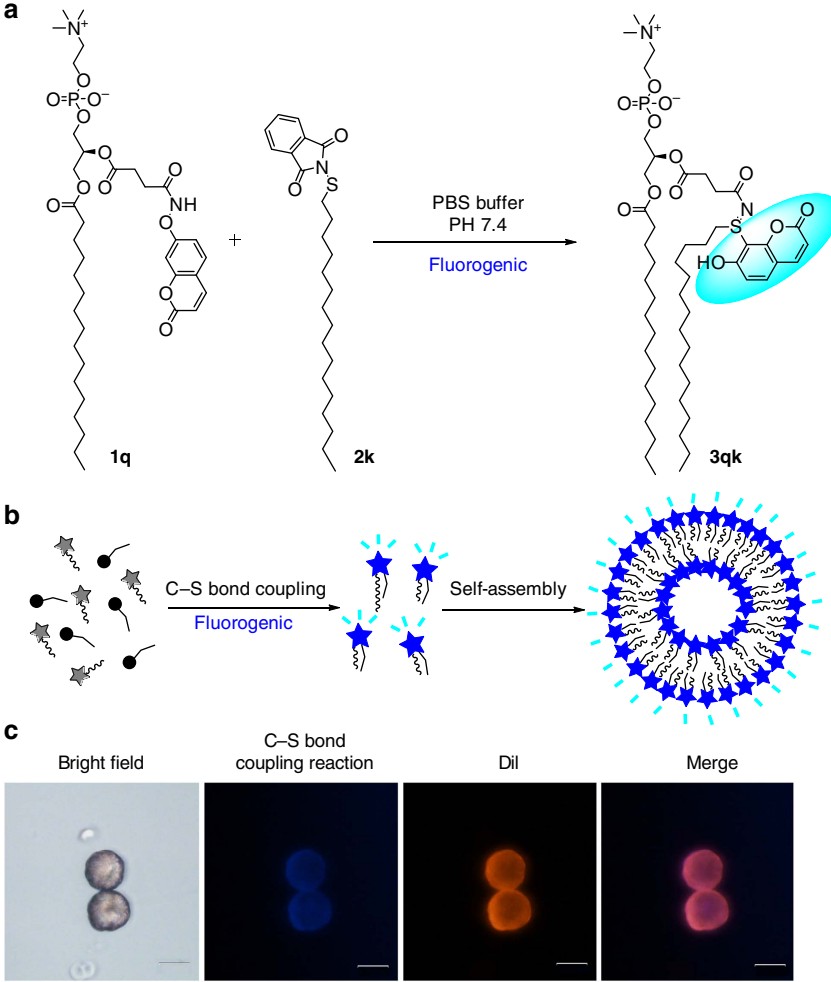

**Figure 4 | Synthesis of fluorogenic phospholipids by C–S bond coupling reaction.** (**a**) Reaction conditions: **1q** (4 mM in PBS buffer, 10 µl) and **2k** (20 mM in CHCl₃, 2 µl) in PBS buffer PH 7.4 (28 µl) was sonicated at RT for 1 h; (**b**) Model of spontaneous fluorescent vesicle assembly induced by C–S bond coupling reaction; (**c**) Fluorescent microscopic images of phospholipid vesicles. Conditions: **1q** (4 mM in PBS buffer, 10 µl) and **2k** (20 mM in CHCl₃, 2 µl) in PBS buffer PH 7.4 (28 µl) was sonicated at RT for 1 h, after 3 h standing at 37 °C, stained with DiI before being imaged on the fluorescence microscopy. Scale bar, 20 µM.

**Data availability**. The X-ray crystallographic coordinates for structures reported in this study have been deposited at the Cambridge Crystallographic Data Centre (CCDC), under deposition numbers CCDC1041436 and CCDC983618. These data can be obtained free of charge from The Cambridge Crystallographic Data Centre via www.ccdc.cam.ac.uk/data_request/cif. The authors declare that all other data supporting the findings of this study are available within the article and Supplementary Information files, and also are available from the corresponding author upon reasonable request.

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

## Acknowledgements

Financial support was provided by the National Science Foundation of China (21622103, 21571098 and 21671099) and the Natural Science Foundation of Jiangsu Province (BK20160022).

## Author contributions

F.X., L.L., Q.W., D.Y. and Y.C. carried out the experimental work; T.-Y.S. and X.Z. carried out the computational work, F.X., L.L., J.J.L and J.Z. wrote the manuscript; J.Z., W.W., Y.L. and W.-Y.S. guided the research.

## Additional information

**Competing interests:** The authors declare no competing financial interests.

**Publisher's note**: 

