## [Peer review file · Nature Communications]

Reviewers' comments:

Reviewer #1 (Remarks to the Author):

In this work, Zhao and colleagues develop a bioinspired ortho-sulfiliminy phenol synthesis using oxyamines and sulfenylation reagents. Its a clever strategy and the authors have thoroughly explored and described reaction scope. Furthermore, they highlight some interesting applications, including the modification of substrates in bacterial lysates and the synthesis of phospholipid vesicles. Their fluorogenic scheme is very interesting and adds impact to work as there is tremendous interest in coupling methods that lead to fluorogenic products. While they demonstrate their method for turning on a coumarin, the general scheme of unmasking a phenol function group should also be applicable to the turn-on of other fluorophores such as fluoresceins etc. Overall, this work should be of interest to a general audience.

Reviewer #2 (Remarks to the Author):

Zhao and co-workers report the metal-free synthesis of ortho sulfilimine substituted phenols via a very interesting rearrangement of N- phenoxy acetamides that is triggered by a sulfenylating agent (a source of S+). The work builds nicely upon the authors previous reports of reactions of N- phenoxy acetamides (refs 31-33). The manuscript begins with the optimisation and discovery of the best sulfenylating agent. Following this, the scope of the reaction is assessed. The scope of the N- phenoxy acetamides is reasonable, with the reaction being amenable to variation in the phenoxy and amide moieties. The phenoxy portion can contain weakly electron donating alkyl groups, electron withdrawing esters, and halides substitution. However, a key example is missing: an electron-donating group such as OMe. I ask the authors to try to add one more example to the scope with OMe substituted in the phenoxy portion. When phenoxy moiety is meta substituted, i.e two inequivalent ortho CHs, then 1:1 mixtures result. The scope with respect to the group introduced through the sulfenylating agent has apparently been investigated based on the SI and manuscript text but Table 2 is missing and instead Table 1 appears again.

The authors' ambitions in applying the method are particularly commendable, and they nicely demonstrate the robustness of their approach by applying the reaction under biocompatible conditions and in the synthesis of a fluorescent phospholipid. This makes the paper of interest to a wider audience, for example in chemical biology.

The reaction contains a very interesting rearrangement and is unusual (a good thing). Therefore I feel that the authors should comment on the mechanism with reference to the literature and by performing some simple experiments: 1) can intermediate B be isolated/observed to prove its structure?; and 2) what results does a cross-over experiment yield between differently substituted intermediates B. This will shed light on the mechanism.

The SI file is in good shape.

For these reasons, the paper is certainly suitable for publication in Nature Communications after the additional changes are also made:

Page 1, line 13. "sulfenylation formation" – rewrite.

Page 1, line 16. "redox versatility", "imine-donating capacity", "cleavability" – what do these phrases mean? I would suggest removing them.

Page 1, line 21. A compound number (1o) should not be mentioned in the abstract.

Page 2, line 29. "sulphur" should be "sulfur"

Page 2. I see only a tenuous link between the biological process in Fig 1a and that discovered by the authors. One appears to be a radical process involving iron while the other involves a sulfur electrophile and no metal. I am not sure this comparison adds anything (other than confusion) to the manuscript. I would suggest that this discussion be removed.

Page 2, line 41. "sulphur" should be "sulfur"

Page 2, line 45. "...may undergo an electrophilic addition to the ortho-position of the X-N directing group" - this needs to be rewritten. The directing group does not have a benzene ring.

Page 3. Figure 1. I don't understand the iron(III) intermediates in (a) - should some bonds to iron be dashed? - and the arrow push that leads to product. I also don't understand the arrow push in (b). This figure should be redrawn.

Page 3, line 61 - "previously reported metal catalyzed conditions"?

Page 3, line 64. I don't think it is necessary to comment on how products were characterized in the main text (although mention of the S=N bond is fine)

Page 4. In Figure 2, please use another abbreviation for the leaving group on sulfur: M is normally used to represent a metal.

Page 4. In Figure 2, (a) the structure of 3aa must be given here. (b) a compound number for the product must be given.

Page 4, line 74. "Tolyl sulfides"?

Page 5, Table 1. (a) not strictly necessary as covered in the main scheme.

Page 6 lines 94-107. As structures 1 are not shown in all cases I think it is best to refer to product structures 3 when talking through the scope and FG tolerance.

Page 6, line 95. "electron-withdrawing groups"

Page 6, line 105. I don't think it is necessary to comment on how products were characterized in the main text.

Page 7. "Table 2" is a copy of the scope from Table 1. The scope with regards to the sulfenylating agent sounds OK.

Page 8. Figure 3. Why does the coumarin have a letter rather than a compound number?

Page 9. Figure 4. Font size is too large in the atom labels for the compounds.

Page 10, line 148. Please check the name of 1p.

Page 10, line 156. "stained"? rewrite please.

Pages 13,14. The references focus on metal-catalyzed processes. I was surprised that the authors do not cite arguably more relevant work on directed, metal-free, redox-neutral, ortho-functionalization. For example, the recent work of Maulide, Procter, Yorimitsu, Peng that involves similar "catch and delivery by rearrangement" approaches. I strongly suggest the authors consider citing this work.

This is excellent work but there are some minor deficiencies in the way the work is presented.

Detailed Responses to Referee's Comments

Reviewers' comments:

Reviewer #1 (Remarks to the Author):

In this work, Zhao and colleagues develop a bioinspired ortho-sulfiliminy phenol synthesis using oxyamines and sulfenylation reagents. It's a clever strategy and the authors have thoroughly explored and described reaction scope. Furthermore, they highlight some interesting applications, including the modification of substrates in bacterial lysates and the synthesis of phospholipid vesicles. Their fluorogenic scheme is very interesting and adds impact to work as there is tremendous interest in coupling methods that lead to fluorogenic products. While they demonstrate their method for turning on a coumarin, the general scheme of unmasking a phenol function group should also be applicable to the turn-on of other fluorophores such as fluoresceins etc. Overall, this work should be of interest to a general audience.

Response

We thank the reviewer for this nice comment.

Reviewer #2 (Remarks to the Author):

Zhao and co-workers report the metal-free synthesis of ortho sulfilimine substituted phenols via a very interesting rearrangement of N-phenoxy acetamides that is triggered by a sulfenylating agent (a source of S⁺). The work builds nicely upon the authors previous reports of reactions of N-phenoxy acetamides (refs 31-33). The manuscript begins with the optimisation and discovery of the best sulfenylating agent. Following this, the scope of the reaction is assessed. The scope of the N-phenoxy acetamides is reasonable, with the reaction being amenable to variation in the phenoxy and amide moieties. The phenoxy portion can contain weakly electron donating alkyl groups, electron withdrawing esters, and halides substitution. However, a key example is missing: an electron-donating group such as OMe. I ask the authors to try to add one more example to the scope with OMe substituted in the

phenoxy portion. When phenoxy moiety is meta substituted, i.e two inequivalent ortho CHs, then 1:1 mixtures result.

The scope with respect to the group introduced through the sulfenylating agent has apparently been investigated based on the SI and manuscript text but Table 2 is missing and instead Table 1 appears again.

The authors' ambitions in applying the method are particularly commendable, and they nicely demonstrate the robustness of their approach by applying the reaction under biocompatible conditions and in the synthesis of a fluorescent phospholipid. This makes the paper of interest to a wider audience, for example in chemical biology.

The reaction contains a very interesting rearrangement and is unusual (a good thing). Therefore I feel that the authors should comment on the mechanism with reference to the literature and by performing some simple experiments: 1) can intermediate B be isolated/observed to prove its structure?; and 2) what results does a cross-over experiment yield between differently substituted intermediates B. This will shed light on the mechanism.

The SI file is in good shape.

For these reasons, the paper is certainly suitable for publication in Nature Communications after the additional changes are also made:

Question 1:

However, a key example is missing: an electron-donating group such as OMe. I ask the authors to try to add one more example to the scope with OMe substituted in the phenoxy portion.

Response

We thank the reviewer for this valuable advice. We have subjected the 4-OMe substituted *N*-phenoxyacetamide to the C–S coupling reaction, the desired product was obtained in 88% yield. The corresponding product was added to Table 1 as compound **31a**.

Revisions Made

Table 1. Substrate scope of aryloxyamides ^[a]

[a] Reaction conditions: 0.2 mmol oxyamide, *N*-ethylthiophthalimides (1.2 equiv) and CsOAc (0.5 equiv) in MeOH (1 mL) at room temperature for 3h. Yields are those of isolated products.

Question 2:

The scope with respect to the group introduced through the sulfenylating agent has apparently been investigated based on the SI and manuscript text but Table 2 is missing and instead Table 1 appears again.

Page 7. "Table 2" is a copy of the scope from Table 1. The scope with regards to the sulfenylating agent sounds OK.

Response

We thank the reviewer for this kind suggestion. We have now added the correct Table 2 in the manuscript.

Revisions Made

Table 2: Substrate scope of *N*-substituted thiophthalimides ^[a]

[a] Reaction conditions: 0.2 mmol **1a**, *N*-substituted thiophthalimides (1.2 equiv) and CsOAc (0.5 equiv) in MeOH (1 mL) at room temperature for 3h. Yields are those of isolated products.

Question 3:

The reaction contains a very interesting rearrangement and is unusual (a good thing). Therefore I feel that the authors should comment on the mechanism with reference to the literature and by performing some simple experiments: 1) can intermediate B be isolated/observed to prove its structure?; and 2) what results does a cross-over experiment yield between differently substituted intermediates B. This will shed light on the mechanism.

Response

We thank the reviewer for this highly valuable advice. To isolate or observe the intermediate B, we made the following efforts.

First, when *N*-(2,6-dimethylphenoxy)acetamide was treated with reagent **2a** under room temperature or 0 °C, intermediate B was never detected (equation a and b). When *S*-*tert*-butyl substituted thiophthalimide **2l** was used as the sulfenylation reagent in the standard condition, the sulfiliminy phenol product was not observed. The reaction between **1a** and **2l** did not proceed (equation c).

Second, we carried out the cross-over experiments using a 1:1 mixture of *N*-phenoxyacetamide **1a** and its analogue **1a-d₈** under the standard condition, only the intramolecular rearrangement products **3aa** and **3aa-d₇** were obtained (Supplementary Fig. 36a), suggesting an intramolecular process. The cross-over experiment between **1c**, **1d** and **2a** confirmed this conclusion (Supplementary Fig. 36b).

Third, the potential energy surface (PES) of the proposed pathway was calculated with density functional theory (DFT), and is depicted in Supplementary Fig. 37a. The computational results suggested that the reaction process containing two main stages: (1) *N*-sulfenylation. (2) [2, 3] sigmatropic rearrangement and aromatization.

Revisions Made

(Please refer to the page 5, paragraph 10 and Supplementary discussion)

Page 5, paragraph 10:

Mechanistic investigation. A combined experimental/computational study was conducted to investigate the reaction mechanism. The cross-over experiment was carried out using a 1:1 mixture of *N*-phenoxyacetamide **1a** and its analogue **1a-d₈** under the standard conditions, only the intramolecular rearrangement products **3aa** and **3aa-d₇** were obtained (Supplementary Fig. 36a), suggesting an intramolecular

process. The cross-over experiment between **1c**, **1d** and **2a** confirmed this conclusion (Supplementary Fig. 36b). To further probe the reaction mechanism, the potential energy surface (PES) of the proposed pathway was calculated with density functional theory (DFT). The computational results suggested that the reaction proceeds through *N*-sulfenylation, [2, 3] sigmatropic rearrangement and aromatization (Supplementary Fig. 37a).

Supplementary discussion:

1. Cross-over Experiment

Supplementary Figure 36. Cross-over experiments. (a) Reaction conditions: **1a** (0.02 mmol), **1a-d₈** (1 eq.), **2a** (2.4 eq.) and CsOAc (1 eq.) in MeOH at room temperature for 3 h. (b) Reaction conditions: **1c** (0.1 mmol), **1d** (1 eq.), **2a** (2.4 eq.)

and CsOAc (1 eq.) in MeOH at room temperature for 3 h. (c) HR-MS spectrum of (a). (d) HR-MS spectrum of (b).

2. DFT Calculations

Computational details

All the calculations were performed with the Gaussian09 suite of programs.⁵ Geometry optimization and energy calculations were conducted with B3LYP.⁶ LANL2DZ + d (0.289) basis set with ECP was used for I and 6-31G (d) basis set was used for atoms.⁷⁻⁹ To verify the stationary points as minima or transition states, vibration frequency calculation at the same level of theory was performed for each structure. The zero-point energy, thermal energy, entropy, and free energy were also derived from vibration frequency. Single point energies were calculated at the B3LYP-D¹⁰/SDD¹¹-6-311++G (d, p) level. A solvent correction for methanol at 298 K was calculated by using SMD solvation model.¹²

As shown in Fig. 2a, for the internal oxidants containing the X–N bond (X=N, S, O), only when X is O, the desired phenol product can be obtained. When N–H was replaced by N–Me, no reaction occurred. Furthermore, base (such as CsOAc) was an indispensable component for this reaction (Supplementary Table 6). These results indicated that the acidity of the N–H is important for this reaction and the N–H bond might be deprotonated by the base to initiate the following reaction.¹³

The *N*-sulfenylation step (**TS1**) was found to be the rate-determining step (RDS). By examining the structure of **TS1**, we envisioned that the energy barrier of nucleophilic substitution process could be influenced by the steric hindrance of the *N*-substituted thiophthalimides. DFT calculation revealed the activation free energy of the reaction between **INT1** and *N*-*t*-butylthiophthalimide was indeed 8 kcal/mol higher (**TS1'**) than that of *N*-ethylthiophthalimide, which implied that *N*-sulfenylation was unlikely to occur for the bulky *N*-*t*-butylthiophthalimide. The experiment was then conducted and confirmed the computational results (Supplementary Fig. 37b).

The calculation results speculated that the reaction process containing two main stages: (1) *N*-sulfenylation. (2) [2, 3] sigmatropic rearrangement and aromatization.

First, the amide (NH) of the substrate could be deprotonated under weak basic conditions such as OAc^- to form the intermediate **INT1**. Next, nucleophilic substitution by the intermediate **INT1** to *N*-ethylthiophthalimide led to the formation of the N–S bond (intermediate **INT2**) via **TS1**, and the computational results indicated that the barrier (25.7 kcal/mol) was accessible under reaction condition. Subsequently, [2,3]-sigmatropic rearrangement occurred through **TS2** to form the intermediate **INT3**. Finally, the intermediate **INT3** would easily tautomerize to the product. (Supplementary Fig. 37a)

Supplementary Figure 37. Mechanistic studies. (a) Free energy (enthalpy) profile of the cascade reaction (in kcal/mol). (b) The reaction of *N*-phenoxyacetamide **1a** and *N*-tert-butylthiophthalimide **2l** under the standard conditions.

Question 4:

Page 1, line 13. “sulfenylation formation” – rewrite.

Response

We thank the reviewer for this suggestion. We have replaced sulfenylation formation with C–S bond formation and sulfoxidation.

Revisions Made

(Please refer to the abstract)

Inspired by an ergothioneine biosynthesis protein EgtB, a mononuclear non-heme iron enzyme capable of catalyzing the ~~sulfenylation formation~~ C–S bond formation and sulfoxidation, we discovered a mild and metal-free C–H sulfenylation/intramolecular rearrangement cascade reaction employing an internally oxidizing O–N bond as a directing group.

Question 5:

Page 1, line 16. “redox versatility”, “imine-donating capacity”, “cleavability” – what do these phrases mean? I would suggest removing them.

Response

We thank the reviewer for this valuable advice. We have removed those phrases and made the following revision.

Revisions Made

(Please refer to the abstract)

Our strategy accommodates a variety of oxyamines with good site selectivity, ~~redox versatility, imine-donating capacity, and easy cleavability.~~ and intrinsic oxidative property.

Question 6:

Page 1, line 21. A compound number (1o) should not be mentioned in the abstract.

Response

We thank the reviewer for this valuable suggestion. We have removed the compound number 1o.

Revisions Made

(Please refer to the abstract)

We demonstrated the biocompatibility of the C–S bond coupling reaction by applying a coumarin-based fluorogenic probe ~~1o~~ in bacterial cell lysates.

Question 7:

Page 2, line 29. “sulphur” should be “sulfur”

Page 2, line 41. “sulphur” should be “sulfur”

Page 3, line 61 – “previously reported metal catalyzed conditions”?

Page 4, line 74. “Tolyl sulfides”?

Page 6, line 95. “electron-withdrawing groups”

Page 10, line 148. Please check the name of 1p.

Page 10, line 156. “stained”? rewrite please.

Response

We thank the reviewer for those valuable advices. We have corrected those spelling mistakes and made the following revisions.

Revisions Made

Page 1, paragraph 1. The key step in its biosynthesis pathway is the mononuclear non-heme iron enzyme EgtB-catalyzed sulfenylation formation between γ -glutamyl cysteine and *N*- α -trimethyl histidine (TMH), involving a ~~sulphur~~ sulfur transfer step and an oxygen transfer step (Fig. 1a).

Page 2, paragraph 3. (i) ~~sulphur~~ sulfur transfer.

Page 2, paragraph 4. At the outset of this study, compounds **1** with those bonds were firstly screened to couple with a thionating reagent *N*-ethylthiophthalimide **2a** under previously reported metal catalyzed conditions for similar reactions (Fig. 2a).

Page 3, paragraph 5. Cresyl Tolyl sulfides with different leaving groups on the *S*-atom such as chloride, tosyl, and phthalimidoyl coupled with *N*-phenoxyacetamide **1a** to afford **3ae** in 18%, 33%, and 85% yield, respectively.

Page 3, paragraph 6. Electron-donating groups, (**1d**, **1e**, **1g**, **1m**), electron-~~withdraw~~ withdrawing groups (**1f**), as well as halogen groups (**1h~1l**) were well tolerated, which afforded substituted sulfilimines in 85% to 92% yield.

Page 4, paragraph 9. We designed a non-fluorescent coumarin-functionalized analogue of the lysolipid 1-palmitoyl-sn-glycero-3-phos-phocholine **1p** and a linear alkyl sulfenylation reagent **2k**.

Page 4, paragraph 9. We confirmed these vesicles were lipid membrane structures by ~~stained~~ staining with the membrane-staining dye 1,1'-dioctadecyl-3,3,3',3'-tetramethylindocarbocyanine perchlorate (DiI)

Question 8:

Page 2. I see only a tenuous link between the biological process in Fig 1a and that discovered by the authors. One appears to be a radical process involving iron while the other involves a sulfur electrophile and no metal. I am not sure this comparison adds anything (other than confusion) to the manuscript. I would suggest that this discussion be removed.

Response

We thank the reviewer for this valuable advice. We wanted to show that these are parallel bond formations: both processes have C–S bond formation and S=X bond

formation. Here we kindly ask the reviewer to allow us to leave the comparison between Figure 1a and 1b since it was the inspiration of our work. However, if the reviewer insists, we will remove this comparison.

Question 9:

Page 2, line 45. "...may undergo an electrophilic addition to the ortho-position of the X-N directing group" – this needs to be rewritten. The directing group does not have a benzene ring.

Response

We thank the reviewer for this valuable advice. We have made the following revision.

Revisions Made

(Please refer to page 2, paragraph 3)

Inspired by the sulfur transferases and our previous successes in O–N bond-directed synthesis of *ortho*-functionalized phenol³³⁻³⁵, we envisioned that *ortho*-sulfiliminy phenols could be obtained by combining a directing group containing an internally oxidizing O–N bond with a sulfenylation reagent.^{36, 37} The desired sulfenylation reagent and oxidizing X–N bond needs to accomplish the following two tasks (Fig. 1b): (i) sulfur transfer.^{38, 39} A well-chosen electrophilic sulfenylation reagent would facilitate the *N*-sulfenylation of the X–N moiety and lead to the formation of an N–S bond to produce intermediate **B**; (ii) ~~simultaneous formation of a C–S bond and an S=N bond. As reported by Billard³⁸ and Ngai,³⁹ the *N*-alkylthio (–NSR) moiety may undergo an electrophilic addition to the *ortho* position of the X–N directing group, leading to the formation of a C–S bond with concurrent X–N bond cleavage, affording the product **C**.~~ rearrangement. Pivotal progress was made by Maulide^{40, 41}, Procter^{32, 42}, Yorimitsu³¹ and Peng⁴³ who pioneered the directed, metal-free, redox-neutral and *ortho*-functionalization. These inspiring work suggested that when the substrate captured a suitable partner, the resulting intermediate may undergo a sigmatropic rearrangement and rearomatization to product **D**, leading to the formation of a C–X

bond with concurrent O–X bond cleavage. Herein, we report a rationally designed and metal-free coupling method to synthesize sulfilimines via an internal oxidant-directing strategy for the cascade formation of C–S and S=N bonds at room temperature.

Question 10:

Page 3. Figure 1. I don't understand the iron(III) intermediates in (a) – should some bonds to iron be dashed? - and the arrow push that leads to product. I also don't understand the arrow push in (b). This figure should be redrawn.

Response

We thank the reviewer for this valuable advice. The bond between iron and nitrogen is dashed line (a coordination bond). We have redrawn the Figure 1 and made the following revision.

Revisions Made

a Enzymatic oxidative C-S bond formation and sulfoxidation

b our work:

Figure 1. Strategy for the formation of *ortho*-sulfiliminy phenol derivatives.

Question 11:

Page 3, line 64. I don't think it is necessary to comment on how products were characterized in the main text (although mention of the S=N bond is fine)

Page 6, line 105. I don't think it is necessary to comment on how products were characterized in the main text.

Response

We thank the reviewer for this valuable advice. We have removed the comments on how products were characterized and made the following revision.

Revisions Made

(Please refer to page 3, paragraph 4; page 4, paragraph 7)

~~The structure of **3aa** was confirmed by NMR spectroscopy, HRMS and X ray crystallography (Supplementary Information, Fig. 30). A bond length of 1.67Å is characteristic of the S=N bond.⁴⁷~~

~~The structure of **3ab** was confirmed by NMR spectroscopy, HRMS and X ray crystallography (Supplementary, Fig. 31).~~

Question 12:

Page 4. In Figure 2, please use another abbreviation for the leaving group on sulfur: M is normally used to represent a metal.

Page 4. In Figure 2, (a) the structure of 3aa must be given here. (b) a compound number for the product must be given.

Response

We thank the reviewer for this valuable advice. We have replaced the M with R as the abbreviation for the leaving group, and showed the structure of **3aa** in Figure 2a, added the compound number **3af** in Figure 2b. The revision is as follows.

Revisions Made

Figure 2. (a) Screening of the multifunctional X-N functional group; Reaction conditions: 0.2 mmol substrate **1**, *N*-ethylthiophthalimides (1.2 equiv), $[\text{Cp}^*\text{RhCl}_2]_2$ (5 mol %) and CsOAc (0.3 equiv) in CH_3CN (1 mL) at room temperature under N_2 for 15h. (b) Screening of different thionating reagents with *N*-phenoxyamides. Reaction conditions: 0.2 mmol substrate **1a**, **2** (1.2 equiv) and CsOAc (0.3 equiv) in MeOH (1 mL) at room temperature for 15h. Yields are those of isolated products. N.R.= No reaction.

Question 13:

Page 5, Table 1. (a) not strictly necessary as covered in the main scheme.

Response

We thank the reviewer for this valuable advice. We have removed the Table 1a from the main scheme.

Revisions Made

Table 1. Substrate scope of aryloxyamides^[a]

[a] Reaction conditions: 0.2 mmol oxamide, *N*-ethylthiophthalimides (1.2 equiv) and CsOAc (0.5 equiv) in MeOH (1 mL) at room temperature for 3h. Yields are those of isolated products.

Question 14:

Page 6 lines 94-107. As structures 1 are not shown in all cases I think it is best to refer to product structures 3 when talking through the scope and FG tolerance.

Response

We thank the reviewer for this valuable advice. We have made the following revision.

Revisions Made

(Please refer to page 3 and 4, paragraph 6 and 7)

To probe the scope of the transition metal-free cascade C–S and S=N bond formation, we examined a series of oxamide substrates (Table 1). Replacing the acetyl group with a bulkier pivaloyl (**Hb**) or a benzoyl (**He**) group only slightly decreased the yield

to 80% (**3ba**) and 83% (**3ca**), respectively. It is worth noting that ~~for substrate 1e,~~ the sulfilimine substitution occurred exclusively at the *ortho*-position of the phenoxyamide moiety instead of the benzamide moiety (**3ca**), which indicated the stronger directing ability of the oxyamide group for sulfenylation. Substitutions on the phenoxy side of **1** had little impact on the yield. Electron-donating groups (~~1d, 1e, 1g, 1m~~ **3da, 3ea, 3ia, 3la**), electron-withdrawing groups (~~1f~~ **3ha**), as well as halogen groups (~~1h-1i~~ **3fa, 3ga**) were well tolerated, which afforded substituted sulfilimines in 85% to 92% yield. The C–S bond formation proceeded exclusively at the site *ortho* to the acetylaminoxy group. Therefore, for substrate **1** with two different *ortho*-sites (~~1g, 1j-1h~~), two regioisomers with ratio almost 1:1 were produced (**3ja:3ja', 3ka:3ka', 3ma:3ma', 3na:3na'**). Fusion of a benzene ring as in the substrate of naphthalene (~~1n~~) did not affect the reaction yield but resulted in high regioselectivity, which only functionalized the *ortho* C–H at C-1 position, resulting in a 2-naphthol derivative (**3oa**).

Under optimal conditions, we explored the substrate scope for *N*-substituted phthalimides (Table 2). The reaction proceeded smoothly for both aliphatic and aromatic thiophthalimides. Aliphatic groups including trifluoromethyl, linear alkyl and cyclic alkyl (~~2a-2d~~) gave high yields (**3ab-3ad**, 76%–92%). For aromatic thiophthalimides, substitutions on the phenyl ring increased the reaction yield (~~2f-2j~~ **2e 3af-3aj > 3ae**). The reaction proceeded well with either electron-donating groups or halogen-containing substrates.

Question 15:

Page 8. Figure 3. Why does the coumarin have a letter rather than a compound number?

Response

We thank the reviewer for this valuable advice. We have replaced the **coumarin P** with the compound number **3pa** and made the following revision.

Revisions Made

Figure 3. (a) C–S bond coupling reaction in aqueous conditions and in the presence of biomolecules. Conditions: **1a** (0.075 mmol), **2a** (0.09 mmol), CsOAc (0.5 equiv), DMSO/PBS buffer = 1:19 (5 mL); The yield was determined by ^1H NMR spectroscopy using 1,4-dimethoxybenzene as an internal standard. The temperature was rt. (b) Reaction of **1p** with **2a** in aqueous conditions. Conditions: **1p** (0.2 mmol), **2a** (0.24 mmol), CsOAc (0.5 equiv), DMSO/PBS buffer = 1:19 (10 mL); Isolated yield. The temperature was rt. (c) Fluorescence spectra of reaction Figure 3b in aqueous conditions. (d) Photograph showing the visual fluorescence of **1p** and coumarin-P **3pa** under a 365 nm UV lamp.

Question 16:

Page 9. Figure 4. Font size is too large in the atom labels for the compounds.

Response

We thank the reviewer for this valuable advice. We have adjusted the font size in the atom labels for the compounds and made the following revision.

Revisions Made

Figure 4. Synthesis of fluorogenic phospholipids by C–S bond coupling reaction.

Question 17:

Pages 13,14. The references focus on metal-catalyzed processes. I was surprised that the authors do not cite arguably more relevant work on directed, metal-free, redox-neutral, ortho-functionalization. For example, the recent work of Maulide, Procter, Yorimitsu, Peng that involves similar “catch and delivery by rearrangement” approaches. I strongly suggest the authors consider citing this work.

Response

We thank the reviewer for this valuable advice. We have cited those work and made the following revision.

Revisions Made

(Please refer to page 2, paragraph 2 and paragraph 3)

A variety of synthetic methods have been developed to construct the *ortho*-functionalized phenols which are highly useful in chemical industry¹⁰, functional materials¹¹ and medicines¹²⁻¹⁴. These methods mainly include three kinds of strategies: (a) Rearrangement of aromatic O–X bonds¹⁵⁻²⁰; (b) directing group-assisted *ortho* C–H hydroxylation of arenes²¹⁻²⁷; and (c) *ortho* C–H functionalization of phenols²⁸⁻³². Although these results have promoted the development of the phenol chemistry, the more efficient, economical and biocompatible methods are still in demand.

31. Yanagi, T. et al. Metal-Free approach to biaryls from phenols and aryl sulfoxides by temporarily sulfur-tethered regioselective C–H/C–H coupling. *J. Am. Chem. Soc.* **138**, 14582-14585 (2016).
32. Shriver, H. J., Fernandez-Salas, J. A., Hedtke, C., Pulis, A.P. & Procter, D.J. Regioselective synthesis of C3 alkylated and arylated benzothiophenes. *Nat. Commun.* **8**, 14801 (2017).

Inspired by the sulfur transferases and our previous successes in O–N bond-directed synthesis of *ortho*-functionalized phenol³³⁻³⁵, we envisioned that *ortho*-sulfiliminyl phenols could be obtained by combining a directing group containing an internally oxidizing O–N bond with a sulfenylation reagent.^{36, 37} The desired sulfenylation reagent and oxidizing X–N bond needs to accomplish the following two tasks (Fig. 1b): (i) sulfur transfer.^{38, 39} A well-chosen electrophilic sulfenylation reagent would facilitate the *N*-sulfenylation of the X–N moiety and lead to the formation of an N–S bond to produce intermediate **B**; (ii) rearrangement. Pivotal progress was made by Maulide^{40, 41}, Procter^{32, 42}, Yorimitsu³¹ and Peng⁴³ who pioneered the directed, metal-free, redox-neutral and *ortho*-functionalization. These inspiring work suggested that when the substrate captured a suitable partner, the resulting intermediate may undergo a sigmatropic rearrangement and rearomatization to product **D**, leading to the formation of a C–X bond with concurrent O–X bond cleavage. Herein, we report a rationally designed and metal-free coupling method to synthesize sulfilimines via an internal oxidant-directing strategy for the cascade formation of C–S and S=N bonds at room temperature.

40. Huang, X., Patil, M., Fares, C., Thiel, W. & Maulide, N. Sulfur(IV)-mediated transformations: from ylide transfer to metal-free arylation of carbonyl compounds. *J. Am. Chem. Soc.* **135**, 7312-7323 (2013).
41. Peng, B., Geerdink, D., Fares, C. & Maulide, N. Chemoselective intermolecular alpha-arylation of amides. *Angew. Chem. Int. Ed.* **53**, 5462-5466 (2014).
42. Fernandez-Salas, J.A., Eberhart, A.J. & Procter, D.J. Metal-Free CH–CH-type cross-coupling of arenes and alkynes directed by a multifunctional sulfoxide group. *J. Am. Chem. Soc.* **138**, 790-793 (2016).
43. Shang, L. et al. Redox-neutral alpha-arylation of alkyl nitriles with aryl sulfoxides: A rapid electrophilic rearrangement. *J. Am. Chem. Soc.* **139**, 4211-4217 (2017).

REVIEWERS' COMMENTS:

Reviewer #2 (Remarks to the Author):

The manuscript has been very carefully revised. I am very happy with the changes. The authors have addressed all the points that I have raised and have carried out all the revisions that I requested. In addition they have carried out the additional mechanistic studies that I suggested and also some computational studies.

I am happy to recommend acceptance of this excellent manuscript.